# Imitating Past Successes can be Very Suboptimal

**Benjamin Eysenbach**[α]   **Soumith Udatha**[α]   **Sergey Levine**[β]   **Ruslan Salakhutdinov**[α]
[α]Carnegie Mellon University      [β]UC Berkeley
beysenba@cs.cmu.edu

## Abstract

Prior work has proposed a simple strategy for reinforcement learning (RL): label experience with the outcomes achieved in that experience, and then imitate the relabeled experience. These outcome-conditioned imitation learning methods are appealing because of their simplicity, strong performance, and close ties with supervised learning. However, it remains unclear how these methods relate to the standard RL objective, reward maximization. In this paper, we formally relate outcome-conditioned imitation learning to reward maximization, drawing a precise relationship between the learned policy and Q-values and explaining the close connections between these methods and prior EM-based policy search methods. This analysis shows that existing outcome-conditioned imitation learning methods do not necessarily improve the policy, but a simple modification results in a method that does guarantee policy improvement, under some assumptions.

## 1   Introduction

Recent work has proposed methods for reducing reinforcement learning (RL) to a supervised learning problem using hindsight relabeling [1, 5, 9, 11, 25, 31]. These approaches label each state action pair with an *outcome* that happened in the future (e.g., reaching a goal), and then learn a conditional policy by treating that action as optimal for making that outcome happen. While most prior work defines outcomes as reaching goal states [5, 9, 25, 28, 31, 37], some work defines outcomes in terms of rewards [3, 14, 30], language [19, 23], or sets of reward functions [7, 16]. We will refer to these methods as *outcome-conditioned behavioral cloning* (OCBC).

OCBC methods are appealing because of their simplicity and strong performance [3, 6, 9]. Their implementation mirrors standard supervised learning problems. These methods do not require learning a value function and can be implemented without any reference to a reward function. Thus, OCBC methods might be preferred over more standard RL algorithms, which are typically challenging to implement correctly and tune [10].

However, these OCBC methods face a major challenge: it remains unclear whether the learned policy is actually optimizing any control objective. Does OCBC correspond to maximizing some reward function? If not, can it be mended such that it does perform reward maximization? Understanding the theoretical underpinnings of OCBC is important for determining how to correctly apply this seemingly-appealing class of methods. It is important for diagnosing why existing implementations can fail to solve some tasks [13, 25], and is important for predicting when these methods will work effectively. Relating OCBC to reward maximization may provide guidance on how to choose tasks and relabeling strategies to maximize a desired reward function.

The aim of this paper is to understand how OCBC methods relate to reward maximization. The key idea in our analysis is to decompose OCBC methods into a two-step procedure: the first step corresponds to averaging together experience collected when attempting many tasks, while the second step reweights the combined experience by task-specific reward functions. This averaging step is equivalent to taking a convex combination of the initial task-conditioned policies, and this averaging

36th Conference on Neural Information Processing Systems (NeurIPS 2022).

can *decrease* performance on some tasks. The reweighting step, which is similar to EM-based policy search, corresponds to policy improvement. While EM-based policy search methods are guaranteed to converge, we prove that OCBC methods can fail to converge because they interleave an averaging step. Because this problem has to do with the underlying distributions, it cannot be solved by using more powerful function approximators or training on larger datasets. While prior work has also presented failure cases for OCBC [25], our analysis provides intuition into when and why these failure cases arise.

The main contribution of our work is an explanation of how outcome-conditioned behavioral cloning relates to reward maximization. We show that outcome-conditioned behavioral cloning does not, in general, maximize performance. In fact, it can result in decreasing performance with successive iterations. However, by appropriately analyzing the conditional probabilities learned via outcome-conditioned behavioral cloning, we show that a simple modification results in a method with guaranteed improvement under some assumptions. Our analysis also provides practical guidance on applying OCBC methods (e.g., should every state be labeled as a success for some task?).

## 2 Preliminaries

We focus on a multi-task MDP with states $s \in \mathcal{S}$, actions $a \in \mathcal{A}^1$, initial state distribution $p_0(s_0)$, and dynamics $p(s_{t+1} \mid s_t, a_t)$. We use a random variable $e \in \mathcal{E}$ to identify each task; $e$ can be either continuous or discrete. For example, tasks might correspond to reaching a goal state, so $\mathcal{E} = \mathcal{S}$. At the start of each episode, a task $e \sim p_e(e)$ is sampled, and the agent receives the task-specific reward function $r_e(s_t, a_t) \in [0, 1]$ in this episode. We discuss the single task setting in Sec. 3.4. We use $\pi(a \mid s, e)$ to denote the policy for achieving outcome $e$. Let $\tau \triangleq (s_0, a_0, s_1, a_1, \cdots)$ be an infinite length trajectory. We overload notation and use $\pi(\tau \mid e)$ as the probability of sampling trajectory $\tau$ from policy $\pi(a \mid s, e)$. The objective and Q-values are:

$$\max_{\pi} \ \mathbb{E}_{p_e(e)}\left[\mathbb{E}_{\pi(\tau|e)}\left[\sum_{t=0}^{\infty} \gamma^t r_e(s_t, a_t)\right]\right], \qquad Q^{\pi(\cdot|\cdot,e)}(s, a, e) \triangleq \mathbb{E}_{\pi(\tau|e)}\left[\sum_{t=0}^{\infty} \gamma^t r_e(s_t, a_t)\Big|_{a_0=a}^{s_0=s}\right]. \quad (1)$$

Because the state and action spaces are the same for all tasks, this multi-task RL problem is equivalent to the multi-objective RL problem [17, 20, 36]. Given a policy $\pi(a \mid s, e)$ and its corresponding Q-values $Q^{\pi(\cdot|\cdot,e)}$, *policy improvement* is a procedure that produces a new policy $\pi'(a \mid s, e)$ with higher average Q-values:

$$\mathbb{E}_{\pi'(\tau|e)}\left[Q^{\pi(\cdot|\cdot,e)}(s, a, e)\right] > \mathbb{E}_{\pi(\tau|e)}\left[Q^{\pi(\cdot|\cdot,e)}(s, a, e)\right] \quad \text{for all states } s.$$

Policy improvement is most often implemented by creating a new policy that acts greedily with respect to the Q-function, but can also be implemented by reweighting the action probabilities of the current policy by (some function of) the Q-values [8, 22, 27]. Policy improvement is guaranteed to yield a policy that gets higher returns than the original policy, provided the original policy is not already optimal [32]. *Policy iteration* alternates between policy improvement and estimating the Q-values for the new policy, and is guaranteed to converge to the reward-maximizing policy [32].

### 2.1 Outcome-Conditioned Behavioral Cloning (OCBC)

Many prior works have used hindsight relabeling to reduce the multi-task RL problem to one of conditional imitation learning. Given a dataset of trajectories, these methods label each trajectory with a task that the trajectory completed, and then perform task-conditioned behavioral cloning [3, 5, 7, 9, 14, 16, 19, 25, 28, 30, 31, 37]. We call this method outcome-conditioned behavioral cloning (OCBC). While OCBC can be described in different ways (see, e.g., [5, 6]), our description below allows us to draw a precise connection between the OCBC problem statement and the standard MDP.

The input to OCBC is a behavioral policy $\beta(a \mid s)$, and the output is an outcome-conditioned policy $\pi(a \mid s, e)$. While the behavioral policy is typically left as an arbitrary choice, in our analysis we will construct this behavioral policy out of task-conditioned behavioral policies $\beta(a \mid s, e)$. This construction will allow us to say whether the new policy $\pi(a \mid s, e)$ is better than the old policy $\beta(a \mid s, e)$. We define $\beta(\tau)$ and $\beta(\tau \mid e)$ as the trajectory probabilities for behavioral policies $\beta(a \mid s)$ and $\beta(a \mid s, e)$, respectively.

---

[1] Most of our analysis applies to both discrete and continuous state and action spaces, with the exception of Lemma 4.1.

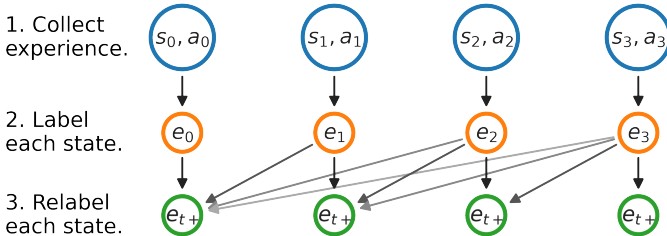

1. Collect experience.
2. Label each state.
3. Relabel each state.

Figure 1: **Labeling and *re*labeling**: *(1)* OCBC collects experience, and then *(2)* labels every state with a task that was completed *at that state*. For example, in goal-reaching problems we have $e_t = s_t$, as state $s_t$ completes the task of reaching goal $s_t$. *(3)* Finally, each state is *relabeled* with a task that was completed *in the future*. Each state may be relabeled with *multiple* tasks, as indicated by the extra green circles.

The key step in OCBC is choosing which experience to use for learning each task, a step we visualize in Fig. 1. After collecting experience with the behavioral policy $\beta(a \mid s)$ (blue circles), *every* state $s_t$ in this experience is labeled with an outcome $e_t$ that is achieved at that state (orange circles). Most commonly, the outcome is equal to the state itself: $e_t = s_t$ [5, 9, 18, 30], though prior work considers other types of outcomes, such as natural language descriptions [19]. We define $p(e_t \mid s_t)$ as the distribution of outcomes achieved at state $s_t$. Finally, we define a random variable $e_{t+}$ to represent the future outcomes for state $s_t$ and action $a_t$. The future outcome is defined as the outcomes for states that occur in the $\gamma$-discounted future. Formally, we can define this distribution over future outcomes as:

$$p^{\beta(a|s)}(e_{t+} \mid s_t, a_t) \triangleq (1 - \gamma)\mathbb{E}_{\beta(\tau)} \left[ \sum_{t=0}^{\infty} \gamma^t p(e_t = e_{t+} \mid s_t) \right]. \tag{2}$$

Outcome-conditioned behavioral cloning then performs conditional behavioral cloning on this relabeled experience. Using $p^{\beta}(s, a)$ to denote the distribution over states and actions visited by the behavioral policy and $p^{\beta}(e_{t+} \mid s, a)$ to denote the distribution over future outcomes (Eq. 2), we can write the conditional behavioral cloning objective as

$$\mathcal{F}(\pi; \beta) = \mathbb{E}_{\substack{p^{\beta}(e_{t+}|s_t,a_t) \\ p^{\beta}(s_t,a_t)}} \left[ \log \pi(a_t \mid s_t, e_{t+}) \right]. \tag{3}$$

---
**Algorithm 1** Outcome-conditioned behavioral cloning (OCBC)

---
**input** behavior policy $\beta(a \mid s)$
$\mathcal{D} \leftarrow \text{RELABEL}(\tau)$ where $\tau \sim \beta(\tau)$ ▷ See Fig. 1.
$\mathcal{F}(\pi; \beta) \leftarrow \mathbb{E}_{(s_t,a_t,e_{t+}) \sim \mathcal{D}} \left[ \log \pi(a_t \mid s_t, e_{t+}) \right]$
$\pi(a \mid s, e) \leftarrow \arg\max_{\pi} \mathcal{F}(\pi)$
**return** $\pi(a \mid s, e)$

---

We define $\pi_O(a \mid s, e) \triangleq \arg\max_{\pi} \mathcal{F}(\pi; \beta)$ as the solution to this optimization problem. We summarize the OCBC training procedure in Alg. 1.

## 3 Relating OCBC to Reward Maximization

In this section, we show that the conventional implementation of OCBC does not quite perform reward maximization. This analysis identifies why OCBC might perform worse, and provides guidance on how to choose tasks and perform relabeling so that OCBC methods perform better. The analysis also allows us to draw an equivalence with prior EM-based policy search methods.

### 3.1 What are OCBC methods *trying* to do?

The OCBC objective (Eq. 3) corresponds to a prediction objective, but we would like to solve a control objective: maximize the probability of achieving outcome $e$. We can write this objective as the likelihood of the desired outcome under the future outcome distribution (Eq. 2) following the task-conditioned policy $\pi(a \mid s, e)$:

$$\mathbb{E}_{e^* \sim p_e(e)} \left[ p^{\pi(\cdot|\cdot,e^*)}(e_{t+} = e^*) \right] = \mathbb{E}_{e^* \sim p_e(e), \pi(\tau|e=e^*)} \left[ \sum_{t=0}^{\infty} \gamma^t \underbrace{(1 - \gamma)p(e_t = e^* \mid s_t)}_{\triangleq r_{e^*}(s_t,a_t)} \right]$$

By substituting the definition of the future outcome distribution (Eq. 2) above, we see that this control objective is equivalent to a standard multi-task RL problem (Eq. 1) with the reward function: $r_e(s_t, a_t) \triangleq (1 - \gamma)p(e_t = e \mid s_t)$. *Do OCBC methods succeed in maximizing this RL objective?*

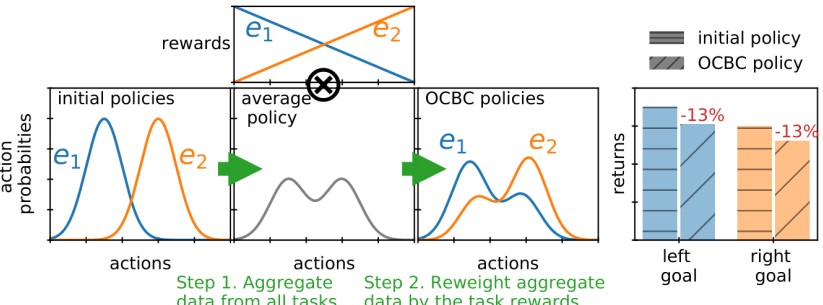

Figure 2: **How does OCBC relate to reward maximization?** OCBC is equivalent to averaging the policies from different tasks *(Step 1)* and then reweighting the action distribution by Q-values *(Step 2)*. The reweighting step is policy improvement, but the averaging step can decrease performance. In this example, OCBC *decreases* the task reward by 13%.

## 3.2 How does OCBC relate to reward maximization?

While it is unclear whether OCBC methods maximize rewards, the OCBC objective is closely related to reward maximizing because the distribution over future outcomes looks like a value function:

**Proposition 3.1.** *The distribution of future outcomes (Eq. 2) is equivalent to a Q-function for the reward function* $r_e(s_t)$*:* $p^{\beta(\cdot|\cdot)}(e_{t+} = e \mid s_t, a_t) = Q^{\beta(\cdot|\cdot)}(s_t, a_t, e)$*.*

This result hints that OCBC methods are close to performing reward maximization. To clarify this connection, we use Bayes' rule to express the OCBC policy $\pi_O(a \mid s, e)$ in terms of this Q-function:

$$\pi_O(a_t \mid s_t, e_{t+}) = p^{\beta(\cdot|\cdot)}(a_t \mid s_t, e_{t+})$$
$$\propto p^{\beta(\cdot|\cdot)}(e_{t+} \mid s_t, a_t)\beta(a_t \mid s_t) = Q^{\beta(\cdot|\cdot)}(s_t, a_t, e = e_{t+})\beta(a_t \mid s_t). \qquad (4)$$

This expression provides a simple explanation for what OCBC methods are doing – they take the behavioral policy's action distribution and reweight it by the Q-values, increasing the likelihood of good actions and decreasing the likelihood of bad actions, as done in prior EM-based policy search methods [22, 26, 27]. Such reweighting is guaranteed to perform policy improvement:

**Proposition 3.2.** *Let* $\pi(a \mid s, e)$ *be the Bayes-optimal policy for applying OCBC to behavioral policy* $\beta(a \mid s)$*. Then* $\pi(a \mid s, e)$ *achieves higher returns than* $\beta(a \mid s)$ *under reward function* $r_e(s_t, a_t)$*:*

$$\mathbb{E}_{\pi(\tau|e)}\left[\sum_{t=0}^{\infty} \gamma^t r_e(s_t, a_t)\right] \geq \mathbb{E}_{\beta(\tau)}\left[\sum_{t=0}^{\infty} \gamma^t r_e(s_t, a_t)\right].$$

The proof is in Appendix C. This initial result is an apples-to-oranges comparison because it compares a task-conditioned policy $\pi(a \mid s, e)$ to a task-agnostic policy, $\beta(a \mid s)$.

To fully understand whether OCBC corresponds to policy improvement, we have to compare successive iterates of OCBC. To do this, assume that data is collected from a task-conditioned behavioral policy $\beta(a \mid s, e)$, where tasks are sampled from the prior $e \sim p_e(e)$.[2] We can now rewrite the Bayes-optimal OCBC policy (Eq. 4) as the output of a two-step procedure: average together the policies for different tasks, and then perform policy improvement:

$$\underbrace{\beta(a \mid s) \leftarrow \int \beta(a \mid s, e)p^\beta(e \mid s)de}_{\text{averaging step}} \qquad \longleftrightarrow \qquad \underbrace{\pi(a \mid s, e) \leftarrow \frac{Q^{\beta(\cdot|\cdot)}(s, a, e)}{p^\beta(e \mid s)}\beta(a \mid s)}_{\text{improvement step}}$$

The distribution $p^\beta(e \mid s)$ is the distribution over *commanded* tasks, given that the agent is in state $s_t$.

Intuitively, the averaging step looks at all the the actions that were taken at state $s_t$, regardless of which task the agent was trying to solve when it took that action. Then, the improvement step reweights those actions, so that actions with high Q-values receive a higher probability and actions with low Q-values receive a lower probability. While it makes sense that the improvement step should yield a better policy, it is less clear whether the averaging step makes the policy better or worse.

---

[2]The state-action distribution $p^\beta(s, a)$ and future outcome distribution $p^\beta(e_{t+} \mid s, a)$ for this *mixture* of policies is equivalent to those for a Markovian policy, $\beta(a \mid s) = \int \beta(a \mid s, e)p^\beta(e \mid s)de$ [38, Thm. 2.8].

### 3.3 OCBC can fail to maximize rewards

We now use this two-step decomposition to show that OCBC can fail to maximize rewards. While prior work has also shown that OCBC can fail [25], our analysis helps to explain why: for certain problems, the averaging step can make the policy much worse. We start by illustrating the averaging and improvement steps with a simple example, and then provide some formal statements using this example as a proof by counterexample.

While priro work has already presented a simple counterexample for OCBC [25], we present a different counterexample to better visualize the averaging and improvement steps. We will use a simple bandit problem with one state, a one-dimensional action distribution, and two tasks. We visualize this setting in Fig. 2. The two tasks have equal probability. The averaging step takes the initial task-conditioned policies *(left)* and combines them into a single policy *(center-bottom)*. Since there is only one state, the weights $p(e \mid s)$ are equal for both tasks. The improvement step reweights the average distribution by the task rewards *(center-top)*, resulting in the final task-conditioned policies learned by OCBC *(right)*. In this example, the final task-conditioned policies achieve returns that are about 13% *worse* than the initial policies.

Formally, this counterexample provides us with a proof of three negative results:

**Proposition 3.3.** *There exists an environment, task distribution, and task $e$ where the OCBC policy, $\pi(a \mid s, e)$, does not achieve the highest rewards:*

$$\mathbb{E}_{\pi(\tau|e)}\left[\sum_{t=0}^{\infty}\gamma^t r_e(s_t, a_t)\right] < \max_{\pi^*}\mathbb{E}_{\pi^*(\tau|e)}\left[\sum_{t=0}^{\infty}\gamma^t r_e(s_t, a_t)\right].$$

**Proposition 3.4.** *There exists an environment, policy $\beta(a \mid s, e)$, and task $e$ such that the policy produced by OCBC, $\pi(a \mid s, e)$, achieves lower returns than the behavior policy:*

$$\mathbb{E}_{\pi(\tau|e)}\left[\sum_{t=0}^{\infty}\gamma^t r_e(s_t, a_t)\right] < \mathbb{E}_{\beta(\tau|e)}\left[\sum_{t=0}^{\infty}\gamma^t r_e(s_t, a_t)\right].$$

Not only can OCBC fail to find the optimal policy, it can yield a policy that is worse than the initial behavior policy. Moreover, iterating OCBC can yield policies that get worse and worse:

**Proposition 3.5.** *There exists an environment, an initial policy $\pi_0(a \mid s, e)$, and task $e$ such that iteratively applying OCBC, $\pi_{t+1} \leftarrow \arg\max_\pi \mathcal{F}(\pi; \beta = \pi_t)$, results in* decreasing *the probability of achieving desired outcomes:*

$$\mathbb{E}_{\pi_{t+1}(\tau|e)}\left[\sum_{t=0}^{\infty}\gamma^t r_e(s_t, a_t)\right] < \mathbb{E}_{\pi_t(\tau|e)}\left[\sum_{t=0}^{\infty}\gamma^t r_e(s_t, a_t)\right] \quad \text{for all } t = 0, 1, \cdots.$$

While prior work has claimed that OCBC is an effective way to learn from suboptimal data [5, 9], this result suggests that this claim is not always true. Moreover, while prior work has argued that *iterating* OCBC is necessary for good performance [9], our analysis suggests that such iteration can actually lead to *worse* performance than doing no iteration.

In this example, the reason OCBC made the policies worse was because of the averaging step. In general, the averaging step may make the policies better or worse. If all the tasks are similar (e.g., correspond to taking similar actions), then the averaging step can decrease the variance in the estimates of the optimal actions. If the tasks visit disjoint sets of states (e.g., the trajectories do not overlap), then the averaging step will have no effect because the averaging coefficients $p^\beta(e \mid s)$ are zero for all but one task. If the tasks have different optimal actions but do visit similar states, then the averaging step can cause performance to decrease. This case is more likely to occur in settings with wide initial state distributions, wide task distributions, and stochastic dynamics. In Sec. 4, we will use this analysis to propose a variant of OCBC that does guarantee policy improvement.

### 3.4 Relationship with EM policy search

While OCBC is conventionally presented as a way to solve control problems without explicit RL [5, 9], we can use the analysis in the preceding sections to show that OCBC has a close connection with

prior RL algorithms based on expectation maximization [4, 12, 15, 21, 22, 26, 27, 33, 34]. We refer to these methods collectively as *EM policy search*. These methods typically perform some sort of reward-weighted behavioral cloning [4, 27], with each iteration corresponding to a policy update of the following form:

$$\max_{\pi} \mathbb{E}_{p^{\beta}(s_t, a_t)} \left[ f(Q^{\beta(\cdot | \cdot)}(s_t, a_t)) \log \pi(a_t \mid s_t) \right],$$

where $f(\cdot)$ is a positive transformation. While most prior work uses an exponential transformation ($f(Q) = e^Q$) [15, 22, 34], we will use a linear transformation ($f(Q) = Q$) [4, 27] to draw a precise connection with OCBC. Prior methods use a variety of techniques to estimate the Q-values, with Monte-Carlo estimates being a common choice [27].

**OCBC is EM policy search.** We now show that each step of EM policy search is equivalent to each step of OCBC, for a particular set of tasks $\mathcal{E}$ and a particular distribution over tasks $p_e(e)$. To reduce EM policy search to OCBC, will use two tasks: task $e_1$ corresponds to reward maximization while task $e_0$ corresponds to reward minimization. Given a reward function $r(s) \in (0, 1)$, we annotate each state with a task by sampling

$$p(e \mid s) = \begin{cases} e_1 & \text{with probability } r(s) \\ e_0 & \text{with probability } (1 - r(s)) \end{cases}.$$

With this construction, the relabeling distribution is equal to the Q-function for reward function $r(s)$: $p^{\beta}(e_{t+} = e_1 \mid s_t, a_t) = Q^{\beta(\cdot | \cdot)}(s_t, a_t)$. We assume that data collection is performed by deterministically sampling the reward maximization task, $e_1$. Under this choice of tasks, the objective for OCBC (Eq. 3) is exactly equivalent to the objective for reward weighted regression [27]:

$$
\begin{aligned}
\mathcal{F}(\pi; \beta) &= \mathbb{E}_{p^{\beta}(s_t, a_t) p^{\beta(\cdot | \cdot)}(e_{t+} | s_t, a_t)} \left[ \log \pi(a_t \mid s_t, e_{t+}) \right] \\
&= \mathbb{E}_{p^{\beta}(s_t, a_t)} \left[ p^{\beta(\cdot | \cdot)}(e_{t+} = e_1 \mid s_t, a_t) \log \pi(a_t \mid s_t, e_1) + p^{\beta(\cdot | \cdot)}(e_{t+} = e_0 \mid s_t, a_t) \log \pi(a_t \mid s_t, e_0) \right] \\
&= \mathbb{E}_{p^{\beta}(s_t, a_t)} \left[ Q^{\beta(\cdot | \cdot)}(s_t, a_t) \log \pi(a_t \mid s_t, e_1) + \cancel{(1 - Q^{\beta(\cdot | \cdot)}(s_t, a_t)) \log \pi(a_t \mid s_t, e_0)} \right].
\end{aligned}
$$

On the last line, we have treated the loss for the reward minimization policy (task $e_0$) as a constant, as EM policy search methods do not learn this policy.

While OCBC can fail to optimize this objective (Sec. 3.2), EM-based policy search methods are guaranteed to optimize expected returns (Dayan and Hinton [4], Toussaint et al. [35, Lemma 1]). This apparent inconsistency is resolved by noting how data are collected. Recall from Sec. 3.2 and Fig. 2 that OCBC is equivalent to an averaging step followed by a policy improvement step. The averaging step emerges because we collect experience for each of the tasks, and then aggregate the experience together. Therein lies the difference: EM policy search methods only collect experience when commanding one task, $\pi(a \mid s, e = e_1)$. Conditioning on a single outcome ($e_1$) during data collection removes the averaging step, leaving just the policy improvement step.

We can apply a similar trick to any OCBC problem: if there is a single outcome that the user cares about ($e^*$), alternate between optimizing the OCBC objective and collecting new experience conditioned on outcome $e^*$. If all updates are computed exactly, such a procedure is guaranteed to yield the optimal policy for outcome $e^*$, maximizing the reward function $r(s) = p(e = e^* \mid s)$.

This trick only works if the user has a single desired outcome. What if the user wants to learn optimal policies for *multiple tasks*? The following section shows how OCBC can be modified to guarantee policy improvement for learning multiple tasks.

## 3.5   Should you relabel with failed outcomes?

While our results show that OCBC may or may not yield a better policy, they also suggest that the choice of tasks $\mathcal{E}$ and the distribution of commanded tasks $p_e(e)$ affect the performance of OCBC. In Appendix A, we discuss whether every state should be labeled as a success for some task, or whether some tasks should be treated as a failure for all tasks. The main result relates this decision to a bias-variance tradeoff: labeling every state as a success for some task decreases variance but potentially incurs bias. Our experiment shows that relabeling some tasks as failures is harmful in the low-data setting, but helpful in settings with large amounts of data.

---

**Algorithm 2** Normalized OCBC

---

$\quad$ **function** RATIOPOLICY$(s, e; \beta(\cdot \mid \cdot, \cdot), \pi(\cdot \mid \cdot, \cdot), \pi_N(\cdot \mid \cdot))$

$\qquad a^{(1)}, a^{(2)}, \cdots \sim \beta(a \mid s, e)$

$\qquad q^{(i)} \leftarrow \frac{\pi(a^{(i)} \mid s, e)}{\pi_N(a^{(i)} \mid s)} \quad$ for $i = 1, 2, \cdots$.

$\qquad i \sim \text{CATEGORICAL}\left( \frac{q^{(1)}}{\sum_i q^{(i)}}, \frac{q^{(2)}}{\sum_i q^{(i)}}, \cdots \right)$

$\qquad$ **return** $a^{(i)}$

$\quad$ **function** POLICYIMPROVEMENT$(\beta(a \mid s, e), \epsilon)$

$\qquad \pi_N(a \mid s) \leftarrow \arg\max_{\pi_N} \mathbb{E}_{\substack{e \sim p_e(e), \tau \sim \beta(\tau \mid e) \\ (s_t, a_t) \sim \tau}} \left[ \log \pi_N(a_t \mid s_t) \right]$

$\qquad \pi(a \mid s, e) \leftarrow \arg\max_{\pi} \mathbb{E}_{\substack{e \sim p_e(e), \tau \sim \beta(\tau \mid e) \\ (s_t, a_t) \sim \tau, k \sim \text{Geom}(1-\gamma)}} \left[ \left( \left| \prod_{t'=t}^{t+k} \frac{\beta(a_{t'} \mid s_{t'}, e_{t+k})}{\beta(a_{t'} \mid s_{t'}, e)} - 1 \right| \leq \epsilon \right) \log \pi(a_t \mid s_t, e_{t+k}) \right]$

$\qquad$ **return** RATIOPOLICY$(s_t, e_{t+}; \beta(\cdot \mid \cdot, \cdot), \pi(\cdot \mid \cdot, \cdot), \pi_N(\cdot \mid \cdot))$

---

# 4 Fixing OCBC

OCBC is not guaranteed to produce optimal (reward-maximizing) policies because of the averaging step. To fix OCBC, we propose a minor modification that "undoes" the averaging step, similar to prior work [25]. To provide a provably-convergent procedure, we will also modify the relabeling procedure, though we find that this modification is unnecessary in practice.

## 4.1 Normalized OCBC

OCBC can fail to maximize rewards because of the averaging step, so we propose a variant of OCBC (*normalized OCBC*) that effectively removes this averaging step, leaving just the improvement step: $\pi(a \mid s, e) \propto Q^{\beta(\cdot \mid \cdot)}(s, a, e) \beta(a \mid s, e)$. To achieve this, normalized OCBC will modify OCBC so that it learns two policies: it learns the task-conditioned policy using standard OCBC (Eq. 3) and additionally learns the average behavioral policy $\pi_N(a \mid s)$ using (unconditional) behavioral cloning. Whereas standard OCBC simply replaces the behavioral policy with the $\pi(a \mid s, e)$, normalized OCBC will *reweight* the behavioral policy by the ratio of these two policies:

$$\tilde{\pi}(a \mid s, e) \leftarrow \frac{\pi(a \mid s, e)}{\pi_N(a \mid s)} \beta(a \mid s, e)$$

The intuition here is that the ratio of these two policies represents a Q-function: if $\pi(a \mid s, e)$ and $\pi_N(a \mid s)$ are learned perfectly (i.e., Bayes' optimal), rearranging Eq. 4 shows that

$$\frac{\pi(a \mid s, e_{t+})}{\pi_N(a \mid s)} \propto Q^{\beta(\cdot \mid \cdot)}(s, a, e = e_{t+})$$

Thus, normalized OCBC can be interpreted as reweighting the policy by the Q-values, akin to EM policy search methods, but without learning an explicit Q-function. This approach is similar to GLAMOR [25], a prior OCBC method that also normalizes the policy by the behavioral policy, but focuses on planning over open-loop action sequences. However, because our analysis relates Q-values to reward maximization, we identify that normalization alone is not enough to prove convergence.

To obtain a provably convergent algorithm, we need one more modification. When relating OCBC to Q-values (Eq. 4), the Q-values reflected the probability of solving task $e$ using *any* policy; that is, they were $Q^{\beta(\cdot \mid \cdot)}(s, a)$, rather than $Q^{\beta(\cdot \mid \cdot, e)}(s, a, e)$. Intuitively, the problem here is that we are using experience collected for one task to learn to solve another task. A similar issue arises when policy gradient methods are used in the off-policy setting (e.g., [29]). While we can correct for this issue using importance weights of the form $\frac{\beta(\tau \mid e)}{\beta(\tau \mid e')}$, such importance weights typically have high variance. Instead, we modify the relabeling procedure. Let us say that the policy for task $e$ collects trajectory $\tau$ which solves task $e'$. We will only use this trajectory as a training example of task $e'$ if the importance weights are sufficiently close to one: $1 - \epsilon \leq \frac{\beta(\tau \mid e')}{\beta(\tau \mid e)} \leq 1 + \epsilon$. If the importance weights are too big or too small, then we discard this trajectory.

The error $\epsilon$ represents a bias-variance tradeoff. When $\epsilon$ is small, we only use experience from one task to solve another task if the corresponding policies are very similar. This limits the amount of data

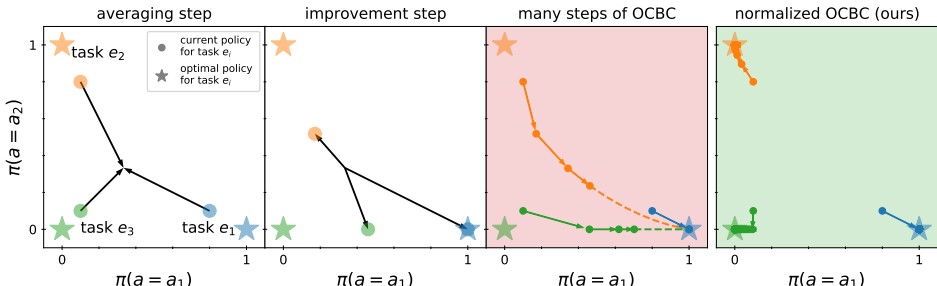

Figure 3: **OCBC = averaging + improvement.** *(Left)* On a bandit problem with three actions and three tasks, we plot the current and optimal policies for each task. OCBC first averages together the action distributions for each task and then *(Left-Center)* learns new policies by reweighting the action distribution by the Q-values for each task. *(Right-Center)* Iterating OCBC (averaging + improvement) produces the optimal policy for task $e_1$ but suboptimal policies for tasks $e_2$ and $e_3$; the final policies for tasks $e_2$ and $e_3$ are *worse* than the initial policies. *(Right)* Normalized OCBC does converge to the optimal policies for each task.

available for training (incurring variance), but minimizes the approximation error. On the other hand, a large value of $\epsilon$ means that experience is shared freely among the tasks, but potentially means that normalized OCBC will converge to a suboptimal policy. Normalized OCBC performs approximate policy iteration, with an approximation term that depends on $\epsilon$ (proof in Appendix C):

**Lemma 4.1.** *Assume that states, actions, and tasks are finite. Assume that normalized OCBC obtains the Bayes' optimal policies at each iteration. Normalized OCBC converges to a near-optimal policy:*

$$\limsup_{k \to \infty} \left| \mathbb{E}_{s_0 \sim p_0(s_0)} \left[ V^{\pi^*(\cdot|\cdot,e)} \right] - \mathbb{E}_{s_0 \sim p_0(s_0)} \left[ V^{\pi_k(\cdot|\cdot,e)} \right] \right| \le \frac{2\gamma}{(1-\gamma)^2} \epsilon.$$

Implementing normalized OCBC is easy, simply involving an additional behavioral cloning objective. Sampling from the ratio policy is also straightforward to do via importance sampling. As shown in Alg. 2, we can first sample many actions from the behavioral policy and estimate their corresponding Q-values using the policy ratio. Then, we select one among these many actions by sampling according to their Q-values. In practice, we find that modifying the relabeling step is unnecessary, and use $\epsilon = \infty$ in our experiments, a decision we ablate in Appendix B.

### 4.2   How to embed reward maximization problems into OCBC problems?

We assume that a set of reward functions is given: $\{r_e(s,a) | e \in \mathcal{E}\}$. Without loss of generality, we assume that all reward functions are positive. Let $r_{\max} \triangleq \max_{s,a,e} r_e(s,a)$ be the maximum reward for any task, at any state and action. We then define an additional, "failure" task: $r_{\text{fail}}(s,a) = r_{\max} - \sum_e r_e(s,a)$. We then use these reward functions to determine how to label each state for OCBC: $p(e_t \mid s_t, a_t) = \frac{1}{r_{\max}} r_e(s_t, a_t)$ for all $e \in \mathcal{E} \cup \{\text{fail}\}$.

## 5   Experiments

Our experiments study three questions. **First**, does OCBC fail to converge to the optimal policy in practice, as predicted by our theory? We test this claim on both a bandit problem and a simple 2D navigation problem. **Second**, on these same tasks, does our proposed fix to OCBC allow the method to converge to the optimal policy, as our theory predicts? Our aim in starting with simple problems is to show that OCBC can fail to converge, *even on extremely simple problems*. **Third**, do these issues with OCBC persist on a suite of more challenging, goal-conditioned RL tasks [9]? We also use these more challenging tasks to understand when existing OCBC methods work well, and when the normalized version is most important. Our experiments aim to understanding when OCBC methods can work and when our proposed normalization can boost performance, not to present an entirely new method that achieves state-of-the-art results. Appendix D contains experimental details. Code to reproduce the didactic experiments in available.[3].

**Bandit problem.**   Our first experiment looks at the learning dynamics of iterated OCBC on a bandit problem. We intentionally choose a simple problem to illustrate that OCBC methods can fail on

---

[3]`https://github.com/ben-eysenbach/normalized-ocbc/blob/main/experiments.ipynb`

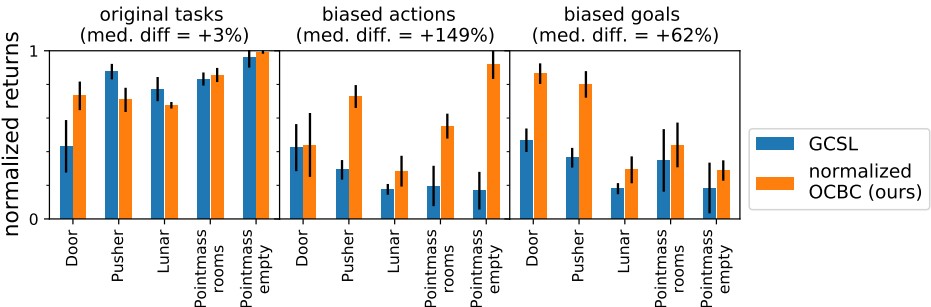

Figure 5: **Goal-conditioned RL benchmark**. We compared normalized OCBC to a prototypical implementation of OCBC (GCSL [9]), using the tasks proposed in that prior work. *(Left)* On the original tasks, both methods perform comparably. On versions of the tasks with *(Center)* skewed action distributions or *(Right)* goal distributions, normalized OCBC performs better, with a median improvement of $\sim 149\%$ and $\sim 62\%$.

even the simplest of problems. The bandit setting allows us to visualize the policy for each task as a distribution over actions. We consider a setting with three actions and three tasks, so the policy for each task can be represented as a point $(\pi(a = a_1 \mid e), \pi(a = a_2 \mid e))$, as shown in Fig. 3. We first visualize the averaging and improvement steps of OCBC. The averaging step *(left)* collapses the three task-conditioned policies into a single, task-agnostic policy. The improvement step *(center-left)* then produces task-conditioned policies by weighting the action distribution by the probability that each action solves the task. The blue task $(e_1)$ is only solved using action $a_1$, so the new policy for task $e_1$ exclusively takes action $a_1$. The green task $(e_3)$ is solved by both action $a_1$ (with probability 0.3) and action $a_3$ (with probability 0.4), so the new policy samples one of these two actions (with higher probability for $a_3$). Iterating OCBC *(center-right)* produces the optimal policy for task $e_1$ but suboptimal policies for tasks $e_2$ and $e_3$; the final policies for tasks $e_2$ and $e_3$ are *worse* than the initial policies. In contrast, normalized OCBC *(right)* does converge to the optimal policies for each task. The fact that OCBC can fail on such an easy task suggests it may fail in other settings.

The bandit setting is a worst-case scenario for OCBC methods because the averaging step computes an *equally*-weighted average of the three task-conditioned policies. In general, different tasks visit different states, and so the averaging step will only average together policies that visit similar states.

**2D navigation.** Our next experiment tests whether OCBC can fail to converge on problems beyond bandit settings. As noted above, we expect the averaging step to have a smaller negative effect in non-bandit settings. To study this question, we use a simple 2D navigation task with two goals (see Appendix Fig. 4 for details). We compare OCBC to normalized OCBC, measuring the average number of steps to reach the commanded goal and plotting results in Fig. 4. At each iteration, we compute the OCBC update (Eq. 3) exactly. While both methods learn near-optimal policies for reaching the blue goal,

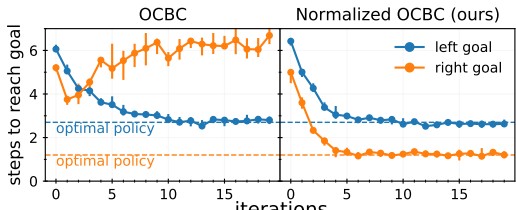

Figure 4: **OCBC $\neq$ policy improvement**: On a tabular environment with two goals, OCBC fails to learn the optimal policy for one of the goals, whereas normalized OCBC learns near-optimal policies for both goals.

only our method learns a near-optimal policy for the orange goal. OCBC's poor performance at reaching the orange goal can be explained by looking at the averaging step. Because the blue goal is commanded nine times more often than the orange goal, the averaging policy $p(a \mid s)$ is skewed towards actions that lead to the blue goal. Removing this averaging step, as done by normalized OCBC, results in learning the optimal policy.

**Comparison on benchmark tasks.** So far, our experiments have confirmed our theoretical predictions that OCBC *can* fail to converge; our analysis does not suggest that OCBC methods will *always* fail. Our next experiments study whether this failure occurs in higher-dimensional tasks. We compare to a recent and prototypical implementation of OCBC, GCSL [9]. To give this baseline a strong footing, we use the goal-reaching benchmark proposed in that paper, reporting *normalized* returns. As shown in Fig. 5 *(Left)*, OCBC and normalized OCBC perform comparably on these tasks. The median improvement from normalized OCBC is a negligible $+3\%$. This result is in line with prior work that finds that OCBC methods can perform well [5, 9, 31], despite our theoretical results.

We hypothesize that OCBC methods will perform worst in settings where the averaging step has the largest effect, and that normalization will be important in these settings. We test this hypothesis by modifying the GCSL tasks in two ways, by biasing the action distribution and by biasing the goal distribution. See Appendix D for details. We show results in Fig. 5. On both sets of imbalanced tasks, our normalized OCBC significantly outperforms OCBC. Normalized OCBC yields a median improvement of $+149\%$ and $+62\%$ for biased settings, supporting our theoretical predictions about the importance of normalization. These results also suggest that existing benchmarks may be accidentally easy for the prior methods because the data distributions are close to uniform, an attribute that may not transfer to real-world problems.

## 6 Conclusion

The aim of this paper is to analyze how outcome-conditioned behavioral cloning relates to training optimal reward-maximizing policies. While prior work has pitched these methods as an attractive alternative to standard RL algorithms, it has remained unclear whether these methods actually optimize any control objective. Understanding how these methods relate to reward maximization is important for predicting when they will work and for diagnosing their failures. Our main result is a connection between OCBC methods and Q-values, a connection that helps relates OCBC methods to many prior methods while also explaining why OCBC can fail to perform policy improvement. Based on our analysis, we propose a simple change that makes OCBC methods perform policy iteration. While the aim of this work was not to propose an entirely new method, simple experiments did show that these simple changes are important for guaranteeing convergence on simple problems.

One limitation of our analysis is that it does not address the effects of function approximation error and sampling error. As such, we showed that the problem with OCBC will persist even when using arbitrarily expressive and tabular policies trained on unlimited data. The practical performance of OCBC methods, and the normalized counterparts that we introduce in this paper, will be affected by these sorts of errors, and analyzing these is a promising direction for future work. A second limitation of our analysis is that it does not focus on how the set of tasks should be selected, and what distribution over tasks should be commanded. Despite these limitations, we believe our analysis may be useful for designing simple RL methods that are guaranteed to maximize returns.

**Acknowledgements.** Thanks to Lisa Lee, Raj Ghugare, and anonymous reviewers for feedback on previous versions of this paper. We thank Keiran Paster for helping us understand the connections with prior work. This material is supported by the Fannie and John Hertz Foundation and the NSF GRFP (DGE1745016).

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
