 

Figure 6: **Relabeling with failures.** *(Left)* On a 1D gridworld, we want to learn policies that move far left or far right. To study the effect of labeling some states as failures, we compare versions of OCBC *(left-center)* where all states are treated as success for the left/right task, versus *(left-bottom)* where some states are treated as failures. We evaluate performance using the average reward *(left-top)*. *(Right)* Labeling some states as failures decreases bias but increases variance: the resulting policies perform worse in the low-data setting but better in the high-data setting.

## A    Should you relabel with failed outcomes?

While our results show that OCBC may or may not yield a better policy, they also suggest that the choice of tasks $\mathcal{E}$ and the distribution of commanded tasks $p_e(e)$ affect the performance of OCBC. When a user wants to apply an OCBC method, they have to choose the set of tasks, which states will be labeled with which tasks, and which tasks will be commanded for data collection. For example, suppose that a human user wants to learn policies for navigating left and right, as shown in Fig. 6. Should every state be labeled as a success for the left or right task, or should some states be labeled as failures for both tasks? If some states are labeled as failures, should the policy be commanded to fail during data collection?

Our two-step decomposition of OCBC provides guidance on this choice. Using a task for data collection means that the policy for that task will be included in the average policy at the next iteration. So one should choose the commanded task distribution so that this average policy is a good *prior* for the tasks that one wants to solve. Assuming that a user does not want to learn a policy for failing, then the task $e_{\text{fail}}$ should not be commanded for data collection.

Even if the failure task is never commanded for data collection, $\pi(a \mid s, e_{\text{fail}})$, should you relabel experience using this failure task? In effect, this question is about how to construct the set of tasks, $\mathcal{E}$. If a task will never be commanded (i.e., $p_e(e) = 0$), then relabeling experience using that task is equivalent to discarding that experience. Indeed, prior work has employed similar procedures for solving reward-maximization problems, discarding experience that receives rewards below some threshold (Oh et al. [24], the 10% BC baseline in Chen et al. [3], Emmons et al. [6]). The decisions to throw away data and the choice of how much data to discard amount to a bias-variance trade off. Discarding a higher percentage of data (i.e., labeling more experience with failure tasks) means that the remaining data corresponds to trials that are more successful. However, discarding data shrinks the total amount of data available for imitation, potentially increasing the variance of the resulting policy estimators. Thus, the amount of experience to relabel with failed outcomes should depend on the initial quantity of data; more data can be labeled as a failure (and hence discarded) if the initial quantity of data is larger.

We demonstrate this effect in a didactic experiment shown in Fig. 6. On a 1-dimensional gridworld, we attempt to learn policies for moving right and left. We consider two choices for labeling: label every state as a success for the left/right task (except the initial state), or label only the outermost states as successes for the left/right task and label all the other states as failures. Fig. 6 *(left)* visualizes this choice, along with the task rewards we will use for evaluation. Labeling many states as failure results in discarding many trials, but increases the success rate of the remaining trials. The results (Fig. 6 *(right)*) show that the decision to include a "failure" task can be useful, if OCBC is provided with enough data. In the low-data regime, including the "failure" task results in discarding too much of the data, decreasing performance.

# B  Filtered Relabeling

Our analysis in Sec. 4 suggested that some experience should not be relabeled with some tasks. Specifically, to prove convergence (Lemma 4.1), normalized OCBC must discard experience if the importance weights $\frac{\beta(\tau|e')}{\beta(\tau|e)}$ are too large or too small. We ran an experiment to study the effect of the clipping parameter, $\epsilon$. For this experiment, we used the "pointmass rooms" task. We ran this experiment twice, once using an initially-random dataset and once using a biased

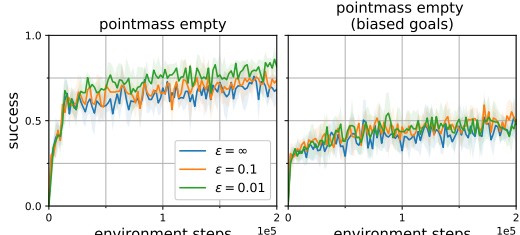

Figure 7: **Filtered relabeling.**

goals dataset (as in Fig. 5). We expected to find that filtered relabeling had a negligible effect when $\epsilon$ was large, but would hurt performance when $\epsilon$ was small (because it reduces the effective dataset size).

We show results in Fig. 7, finding that filtered relabeling does not decrease performance. On the surface, these results are surprising, as they indicate that throwing away many training examples does not decrease performance. However, our theory in Sec. 4 suggests that those transitions with very large or small importance weights may preclude convergence to the optimal policy. So, in the balance, throwing away potentially-harmful examples might be beneficial, even though it decreases the total amount of experience available for training.

# C  Analysis

## C.1  Proof that Reweighting Performs Policy Improvement

**Proposition C.1.** *Let a behavior policy $\beta(a \mid s)$ and reward function $r(s, a)$ be given. Let $Q^\beta$ denote the corresponding Q-function. Define the reweighted policy as*

$$\tilde{\pi}(a \mid s) \triangleq \frac{Q^\beta(a \mid s)\beta(a \mid s)}{\int Q^\beta(a' \mid s)\beta(a' \mid s)da'}.$$

*Then this reweighting step corresponds to policy improvement:*

$$E_{\tilde{\pi}}\left[Q^\beta(s, a)\right] \geq E_\beta\left[Q^\beta(s, a)\right].$$

*Proof.* Let state $s$ be given. We start by substituting the definition of reweighted policy and then applying Jensen's inequality:

$$
\begin{aligned}
E_{\tilde{\pi}}\left[Q^\beta(s, a)\right] &= \int \tilde{\pi}(a \mid s)Q^\beta(s, a)da \\
&= \int \frac{Q^\beta(a \mid s)\beta(a \mid s)}{\int Q^\beta(a' \mid s)\beta(a' \mid s)da'}Q^\beta(s, a)da \\
&= \frac{\int (Q^\beta(s, a))^2\beta(a \mid s)da}{\int Q^\beta(a \mid s)\beta(a \mid s)da} \\
&= \frac{\mathbb{E}_\beta[(Q^\beta(s, a))^2]}{\mathbb{E}_\beta[Q^\beta(s, a)]} \\
&\geq \frac{\mathbb{E}_\beta[Q^\beta(s, a)]^2}{\mathbb{E}_\beta[Q^\beta(s, a)]} \\
&= \mathbb{E}_\beta[Q^\beta(s, a)].
\end{aligned}
$$

$\square$

## C.2  Normalized OCBC performs Policy Improvement

In this section, we prove that normalized OCBC performs policy improvement (Lemma 4.1)

*Proof.* Our proof will proceed in two steps. The first step analyzes Q-values. Normalized OCBC uses Q-values from the average policy, but we need to analyze the task-conditioned Q-values to prove policy improvement. So, in the first step we prove that the Q-values for the average policy and task-conditioned policies are similar. Then, in the second step, we prove that using approximate Q-values for policy iteration yields an approximately-optimal policy. This step is a direct application of prior work [2].

**Step 1.** The first step is to prove that the average Q-values are close to the task-conditioned Q-values. Below, we will use $R_e(\tau) \triangleq \sum_{t=0}^{\infty} \gamma^t r_e(s_t, a_t)$:

$$
\begin{aligned}
\left| Q^{\beta(\cdot|\cdot,e)}(s,a,e) - Q^{\beta(\cdot|\cdot,e')}(s,a,e) \right| &= \left| \int \beta(\tau \mid s,a,e) R_e(\tau) d\tau - \int \beta(\tau \mid s,a,e') R_e(\tau) d\tau \right| \\
&= \left| \int \left( \beta(\tau \mid s,a,e) - \beta(\tau \mid s,a,e') \right) R_e(\tau) d\tau \right| \\
&= \left| \int \beta(\tau \mid s,a,e) \left( 1 - \frac{\beta(\tau \mid s,a,e')}{\beta(\tau \mid s,a,e)} \right) R_e(\tau) d\tau \right| \\
&\leq \int \left| \beta(\tau \mid s,a,e) \left( 1 - \frac{\beta(\tau \mid s,a,e')}{\beta(\tau \mid s,a,e)} \right) \right| d\tau \cdot \max_{\tau} |R_e(\tau) d\tau| \\
&\leq \int \beta(\tau \mid s,a,e) \left| 1 - \frac{\beta(\tau \mid s,a,e')}{\beta(\tau \mid s,a,e)} \right| d\tau \cdot 1 \\
&= \mathbb{E}_{\beta(\tau|s,a,e)} \left[ \left| 1 - \frac{\beta(\tau \mid s,a,e')}{\beta(\tau \mid s,a,e)} \right| \right] \\
&\leq \epsilon.
\end{aligned}
$$

The first inequality is an application of Hölder's inequality. The second inequality comes from the fact that $r_e(s,a) \in [0, 1-\gamma]$, so the discounted some of returns satisfies $R_e(\tau) \in [0,1]$. The final inequality comes from how we perform the relabeling. Thus, we have the following bound on Q-values:

$$
Q^{\beta(\cdot|\cdot,e)}(s,a,e) - \epsilon \leq Q^{\beta(\cdot|\cdot,e')}(s,a,e) \leq Q^{\beta(\cdot|\cdot,e)}(s,a,e) + \epsilon. \tag{5}
$$

Since Q-values for the average policy are a convex combination of these Q-values ($Q^{\beta(\cdot|\cdot)}(s,a) = \int Q^{\beta(\cdot|\cdot,e')}(s,a,e) p^{\beta}(e' \mid s,a) de'$), this inequality also holds for the Q-values of the average policy:

$$
Q^{\beta(\cdot|\cdot,e)}(s,a,e) - \epsilon \leq Q^{\beta(\cdot|\cdot)}(s,a,e) \leq Q^{\beta(\cdot|\cdot,e)}(s,a,e) + \epsilon. \tag{6}
$$

**Step 2.** At this point, we have shown that normalized OCBC is equivalent to performing policy iteration with an approximate Q-function. In the second step, we employ [2, Proposition 6.2] to argue that policy iteration with an approximate Q-function converges to an approximately-optimal policy:

$$
\limsup_{k \to \infty} \| V^{\pi^*(\cdot|\cdot,e)} - V^{\pi_k(\cdot|\cdot,e)} \|_{\infty} \leq \frac{2\gamma}{(1-\gamma)^2} \epsilon. \tag{7}
$$

The L-$\infty$ norm means that this inequality holds for the values of every state. Thus, it also holds for states sampled from the initial state distribution $p_0(s_0)$:

$$
\limsup_{k \to \infty} \left| \mathbb{E}_{s_0 \sim p_0(s_0)} \left[ V^{\pi^*(\cdot|\cdot,e)(s_0)} \right] - \mathbb{E}_{s_0 \sim p_0(s_0)} \left[ V^{\pi_k(\cdot|\cdot,e)}(s_0) \right] \right| \leq \frac{2\gamma}{(1-\gamma)^2} \epsilon. \tag{8}
$$

The value of the initial state is simply the expected return of the policy:

$$
\limsup_{k \to \infty} \left| \mathbb{E}_{\pi^*(\tau|e)} \left[ \sum_{t=0}^{\infty} \gamma^t r_e(s_t, a_t) \right] - \mathbb{E}_{\pi_k(\tau|e)} \left[ \sum_{t=0}^{\infty} \gamma^t r_e(s_t, a_t) \right] \right| \leq \frac{2\gamma}{(1-\gamma)^2} \epsilon. \tag{9}
$$

Rearranging terms, we observe that normalized OCBC converges to an approximately-optimal policy:

$$
\limsup_{k \to \infty} \mathbb{E}_{\pi_k(\tau|e)} \left[ \sum_{t=0}^{\infty} \gamma^t r_e(s_t, a_t) \right] \geq \max_{\pi^*(\cdot|\cdot,e)} \mathbb{E}_{\pi^*(\tau|e)} \left[ \sum_{t=0}^{\infty} \gamma^t r_e(s_t, a_t) \right] - \frac{2\gamma}{(1-\gamma)^2} \epsilon. \tag{10}
$$

$\square$

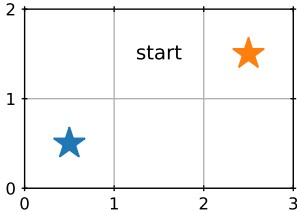

Figure 8: Gridworld environment used for the experiment in Fig. 4

## D   Experimental Details

We have released a Jupyter notebook[4] for reproducing all of the small-scale experiments (Figures 2, 3, 4 and 6). This notebook runs in less than 60 seconds on a standard CPU workstation.

The comparison with GCSL was done by changing a few lines of code in the official GCSL implementation[5] to support training a second (marginal) policy. We used all the default hyperparameters. We ran five random seeds for each of these experiments, and error bars denote the standard deviation.

**Original tasks (Fig. 5 *(Left)*).**   This experiment was run in the online setting, following Ghosh et al. [9]. The pseudocode for these tasks is:

```
replay_buffer = ReplayBuffer()
policy = RandomInitPolicy()
for t in range(num_episodes):
  experience = Collect(policy, original_goal_distribution)
  replay_buffer.extend(experience)
  policy = UpdatePolicy(policy, replay_buffer)
results = Evaluate(policy, original_goal_distribution)
```

For this experiment, we evaluated the policies in two different ways: sampling actions from the learned policy, and taking the argmax from the policy distribution. We found that GCSL performed better when sampling actions, and normalized OCBC performed better when taking the argmax; these are the results that we reported in Fig. 5 *(Left)*. For the biased experiments, we used the "sampling" approach for both GCSL and normalized OCBC.

**Biased actions (Fig. 5 *(Center)*).**   This experiment was run in the offline setting. For data collection, we took a pre-trained GCSL policy and sampled an *suboptimal* action (uniformly at random) with probability 95%; with probability 5%, we sampled the argmax action from the GCSL policy. The pseudocode for these tasks is:

```
replay_buffer = ReplayBuffer()
while not replay_buffer.is_full():
  if env.is_done():
    s = env.reset()
  action_probs = expert_policy(s)
  best_action = argmax(action_probs)
  if random.random() < 0.95:
    a = random([a for a in range(num_actions) if a != best_action])
  else:
    a = best_action
  replay_buffer.append((s, a))
  s = env.step(a)
policy = RandomInitPolicy()
for t in range(num_episodes):
```

---

[4]https://github.com/ben-eysenbach/normalized-ocbc/blob/main/experiments.ipynb
[5]https://github.com/dibyaghosh/gcsl

```
    policy = UpdatePolicy(policy, replay_buffer)
results = Evaluate(policy, original_goal_distribution)
```

**Biased goals** (**Fig. 5** *(Right)*).    This experiment was run in the online setting. We introduced a bias
to the goal space distribution of the environments by changing the way the goal is sampled. We used
this biased goal distribution for both data collection and evaluation. The pseudocode for these tasks
is:

```
policy = RandomInitPolicy()
replay_buffer = ReplayBuffer()
for t in range(num_episodes):
  if not replay_buffer.is_full():
    experience = Collect(policy, biased_goal_distribution)
    replay_buffer.extend(experience)
  policy = UpdatePolicy(policy, replay_buffer)
results = Evaluate(policy, biased_goal_distribution)
```

As the goal distribution for each environment is different, the respective bias introduced is different
for each task:

- **Pusher**: The four dimensional goal includes the $(x, y)$ positions of the hand and the puck.
  Instead of sampling for both positions independently, we just sample the puck-goal position
  and set the hand-goal position to be the same as the puck-goal position.
- **Door**: The 1-dimensional goal indicates the angle to open the door. We bias the goal space
  to just sample the goals from one half of the original goal space.
- **Pointmass Rooms**: The goal is 2-dimensional: the $(x, y)$ position of the goal. We sample the
  $x$ uniformly from $[-0.85, 0.85]$ (the original range is $[-1, 1]$), and then set the $y$ coordinate
  of the goal equal to the $x$ coordinate of the goal.
- **Pointmass Empty**: The goal is 2-dimensional: the $(x, y)$ position of the goal. We sample
  the $x$ coordinate uniformly from $[-0.9, -0.45]$, and set the $y$ coordinate of the goal equal to
  the $x$ coordinate of the goal.
- **Lunar**: The goal is 5-dimensional. We sample the $x$ coordinate (`g[0]`) uniformly from
  [-0.3,0.3]. The $y$ coordinate (`g[1]`) is fixed to 0. The lander angle (`g[2]`) is fixed to its
  respective `g_low`. The last two values in the goal space(`g[3]`, `g[4]`) are fixed to be 1,
  which implies that the ground-contact of legs is `True` at the goal location.