# OpenReview forum: "Imitating Past Successes can be Very Suboptimal"
_NeurIPS.cc/2022/Conference — NeurIPS 2022 Accept_

### Official Review · Reviewer_Z1NT · 2022-07-09

**Rating:** 7
**Confidence:** 4
**Soundness:** 3 good
**Presentation:** 4 excellent
**Contribution:** 2 fair

**Summary:**

This paper discusses about the outcome-conditioned behavior cloning (OCBC) method in the goal-conditioned reinforcement learning field. It suggests that OCBC is not guaranteed to produce optimal (reward-maximizing) policies because of the averaging step (averaging the policies from different tasks). Furthermore, a normalized OCBC is proposed to fix the problem above. Both theoretical and experimental evidences are provided to support the suboptimality of OCBC.

**Questions:**

Is the conclusion and proposed method (normalized OCBC) suitable to single-task behavior cloning, both online and offline settings?
Is OCBC the mainstream method in goal-conditioned RL? Are there any other variants of OCBC?

**Limitations:**

The authors adequately address the limitations.

**Strengths And Weaknesses:**

The discussion of OCBC's suboptimality is novel and interesting. Authors provide solid theoretical analyses and easy-to-understand illustrations. However, it would be better if the significance and related works are more detailedly clarified.

---

> ### Author Response · Authors · 2022-07-29
> **Response to Z1NT**
>
> We thank the reviewer for the time they spent reviewing the paper. Below, we answer the reviewers questions.
>
> > Is the conclusion and proposed method (normalized OCBC) suitable to single-task behavior cloning, both online and offline settings?
>
> No, the main conclusion of the paper (that OCBC can fail to maximize rewards) doesn't apply to single-task settings. Whether single-task behavioral cloning works or not depends on the type of data it is applied to (e.g., random data, expert data).
>
> Regarding the proposed method, yes, normalized OCBC could be applied to the single-task setting. As discussed in Sec. 3.3, OCBC can be applied to single-task settings, and it ends up resembling prior EM policy search methods. Whereas these EM policy search methods update the policy $\tilde{\pi}(a \mid s) \propto Q^\beta(s, a)\beta(a \mid s)$, normalized OCBC would correspond to an update $\tilde{\pi}(a \mid s) \propto Q^\beta(s, a)$. This less regularized update might be faster in some settings, but might be more likely to get stuck in local optima.
>
> > Is OCBC the mainstream method in goal-conditioned RL?
>
> Yes, OCBC is a very mainstream method for goal-conditioned RL, where it is typically called GCBC. Perhaps the other mainstream method for goal-conditioned RL is off-policy RL with HER [1, 2, 3]. OCBC/GCBC is simpler and tends to achieve higher rewards [4], especially in the offline setting [5].
>
> > Are there any other variants of OCBC?
>
> There are many different design decisions, such as whether to quantize actions, how to sample the future states, and the particular choice of architecture. See [5] for a discussion of these decisions. For fair comparison in our experiments, we implemented normalized OCBC using exactly the same hyperparameters as our base OCBC implementation
>
> > it would be better if the significance and related works are more detailedly clarified.
>
> We would be happy to incorporate any specific citations or feedback that the reviewer would like to provide.
>
> --------------------
> [1] Andrychowicz, Marcin, et al. "Hindsight experience replay." Advances in neural information processing systems 30 (2017).
>
> [2] Fang, Meng, et al. "Curriculum-guided hindsight experience replay." Advances in neural information processing systems 32 (2019).
>
> [3] Pong, Vitchyr, et al. "Temporal Difference Models: Model-Free Deep RL for Model-Based Control." International Conference on Learning Representations. 2018.
>
> [4] Ghosh, Dibya, et al. "Learning to reach goals via iterated supervised learning." arXiv preprint arXiv:1912.06088 (2019).
>
> [5] Emmons, Scott, et al. "RvS: What is Essential for Offline RL via Supervised Learning?." arXiv preprint arXiv:2112.10751 (2021).

---

### Official Review · Reviewer_MVgi · 2022-07-10

**Rating:** 6
**Confidence:** 3
**Soundness:** 2 fair
**Presentation:** 3 good
**Contribution:** 2 fair

**Summary:**

This paper studies an existing family of methods for multi-task Reinforcement learning called outcome conditioned behaviour cloning (OCBC). The paper identifies a short-coming of these methods, that, these methods are not guaranteed to maximize rewards when applied to multi-task RL. The paper proposes a correction to these method and shows that the proposed method is guaranteed to obey the policy improvement theorem. The experiments show that the proposed method performs better than OCBC under various settings.

**Questions:**

My main confusion lies in the problem setting and problem formulation.
What is multi-task RL?
How is it different from multi-objective RL?
The paper presents preliminaries and many mathematical definitions regarding the Q values and distributions of trajectories. However, the paper fails to mention what the aim of the agent in this setting is?
Does the agent want to maximize rewards?
Does the agent want to reproduce a trajectory that leads to a state where it achieves a tasks?
What is the relation between task and reward?
What if the original algorithm OCBC was designed for the agent to achieve the task and it does not really care about maximizing an abstract notion of reward?
Why is reward maximization important in this setting
Generally speaking, it is fair to say that goal of almost every MDP agent is reward maximization. Is that not true in this case?

The most simplest experimental setting the paper presents is of a multi-armed bandit. But here again, the paper presets a multi-task multi-armed bandit? What is this setting. Does the actions not have reward associated with each arm? Are the rewards of these arms dependent on each other since each action accomplishes a task? Again, what is the relation between reward and task?


**Ethics Review Area:**

["I don’t know"]

**Limitations:**

The problem setting and formulation for this paper were quite confusing. It is not possible to gather the significance and relevance of the results without a clear understanding of the goal of the agent. The paper will benefit from clearly describing the problem setting and goal. This should naturally answer the question regarding the relationship between reward and multi-tasking.

**Strengths And Weaknesses:**

Strengths
+ Seems to follow and study an existing suite of methods and analyzes their theoretical relevance in depth
+ Follows a good step-wise process towards building its results
+ Presents a theoretical, algorithmic and experimental contribution
Weakness
- The problem setting and formulation was not clear
- The experimental setting was also not clear (related to previous weakness)
- Difficult to understand the significance of these results

---

> ### Author Response · Authors · 2022-07-29
> **Response to MVgi**
>
> We thank the reviewer for their review of the paper, as well as their thoughtful suggestions for improvement. We believe that many of the concerns stem from a misunderstanding of the paper's problem statement, which we will aim to clarify in the paper. We address the specific questions below. Could the reviewer let us know if these answers clarify the issues?
>
> **Problem setting**: We define the agent's objective as reward maximization in Eq. 1. When OCBC methods were originally proposed (e.g., [1, 2, 3]), it was unclear how they related to reward maximization. The main contribution of this paper is to show that OCBC methods do not perform reward maximization in general, and to propose a modification that does allow them to perform reward maximization. Thus, we believe that our work is useful because it allows OCBC methods to be studied under the same MDP reward maximization framework as other RL methods.
>
> **Detailed questions**:
> > What is multi-task RL? How is it different from multi-objective RL?
>
> Multi-task RL [6, 7] is the problem of solving many MDPs $\{\mathcal{M}_e \mid e \in \mathcal{E}\}$. The agent observes the MDP identifies $e$ and selects actions to maximize rewards in $\mathcal{M}_e$. Typically, these MDPs have some shared structure, such as similar dynamics or similar reward functions. Our problem setting is similar to [6], in that the the MDPs all have the same dynamics and differ only in the reward function.
>
> Multi-objective RL [4, 5] typically refers to a single-task RL problem where the single reward function is a combination of multiple reward functions.
>
> > Does the agent want to maximize rewards?
>
> Yes, we focus on maximizing rewards (see Eq. 1).
>
> > Does the agent want to reproduce a trajectory that leads to a state where it achieves a tasks?
>
> This is how most prior work defines the objective for OCBC. One limitation of this definition is that it is unclear how it relates to reward maximization. The main contribution of this paper is to show that OCBC methods *almost* (but not quite) maximize a certain reward function, and could be modified to do so more reliably.
>
> > What is the relation between task and reward?
>
> Each task has a distinct reward function (L52). Another way of saying this is that there are many MDPs, one for each task, and the MDPs are all identical except for their reward functions. This is a common setting in prior work on multi-task RL [6, 8, 9].
>
> > What if the original algorithm OCBC was designed for the agent to achieve the task and it does not really care about maximizing an abstract notion of reward? Why is reward maximization important in this setting?
>
> Evaluating the agent based on whether it achieved the task or not is mathematically equivalent to evaluating it based on the expected reward (L121), for the reward function defined in L115.
> We may have misunderstood this question, and welcome the reviewer to clarify it.
>
> > Generally speaking, it is fair to say that goal of almost every MDP agent is reward maximization. Is that not true in this case?
>
> Yes. One reason we believe the main contribution of this paper (relating OCBC to MDPs and reward maximization) is useful is because MDPs are so widely studied.
>
> > The most simplest experimental setting the paper presents is of a multi-armed bandit. But here again, the paper presets a multi-task multi-armed bandit? What is this setting?
>
> This is a multi-task bandit with continuous actions. The reward functions shown in Fig 2 (top center).
>
> > Does the actions not have reward associated with each arm?
>
> Actions do result in observing the corresponding reward value. When the agent takes action $a$, it observes $r_e(a)$.
>
> > Are the rewards of these arms dependent on each other since each action accomplishes a task?
>
> The rewards are deterministic functions of the action (see Fig 2 (top center)).
>
> **Do these responses help to clarify the reviewer's questions about the problem setting?** What additional questions does the reviewer have about the problem setting, or other aspects of the paper? We look forward to continuing the discussion.

---

> > ### Author Response · Authors · 2022-07-29
> > **Response to MVgi**
> >
> > References:
> >
> > [1] Sun, Hao, et al. "Policy continuation with hindsight inverse dynamics." Advances in Neural Information Processing Systems 32 (2019).
> >
> > [2] Ding, Yiming, et al. "Goal-conditioned imitation learning." Advances in Neural Information Processing Systems 32 (2019).
> >
> > [3] Ghosh, Dibya, et al. "Learning to Reach Goals via Iterated Supervised Learning." International Conference on Learning Representations. 2020.
> >
> > [4] K. Van Moffaert, M. M. Drugan and A. Nowé, "Scalarized multi-objective reinforcement learning: Novel design techniques." IEEE Symposium on Adaptive Dynamic Programming and Reinforcement Learning (ADPRL). 2013.
> >
> > [5] Mossalam, Hossam, et al. "Multi-objective deep reinforcement learning." arXiv preprint arXiv:1610.02707 (2016).
> >
> > [6] Li, Alexander, Lerrel Pinto, and Pieter Abbeel. "Generalized hindsight for reinforcement learning." Advances in neural information processing systems 33 (2020): 7754-7767.
> >
> > [7] Wilson, Aaron, et al. "Multi-task reinforcement learning: a hierarchical bayesian approach." Proceedings of the 24th international conference on Machine learning. 2007.
> >
> > [8] Landolfi, Nicholas C., Garrett Thomas, and Tengyu Ma. "A model-based approach for sample-efficient multi-task reinforcement learning." arXiv preprint arXiv:1907.04964 (2019).
> >
> > [9] Laroche, Romain, and Merwan Barlier. "Transfer reinforcement learning with shared dynamics." Thirty-First AAAI Conference on Artificial Intelligence. 2017.

---

> > ### Comment · Reviewer_MVgi · 2022-08-07
> > **response**
> >
> > You are right, my confusion is in the problem setting. I went back to check the definition of objective in section 2. However, reading it again only adds to my confusion. So we have s state, a action but what are these variable part of (some state space, action space)? Are they integers (discrete)? Are they continuous on real-line?
> > What is e? How does one define a task in MDP if not for the reward function? Why do I need anything other than a reward function to define a task?
> >
> > I read the reference [6] (not completely). The description in paper 6 clears quite a few things. I would request the authors to please write the preliminary section so that the paper is self-contained.
> >
> > I do not see the difference between multi-objective RL and multi-task RL especially if the objective is to maximize expected reward. If the dynamics and states and actions are the same and that multi-task MDPs only differ in the reward then I don’t see how it is different from multi-objective RL. It also has the same definition. (It is not always necessary that the weight combinations of multiple reward functions are available in advance)
> > The weight combination in the case of multi-task RL (as the setting in this paper) is just the probability distribution over tasks.
> >
> > If OCBC defines the objective to produce trajectories that accomplish tasks, that makes sense, since it basically means that OCBC abandons the idea of reward maximization in order for the agent to learn to achieve certain tasks. The idea of maximizing a weighted combination of different reward functions is multi-objective RL. Hence, it is not a surprise that this paper proves that OCBC does not maximize reward (since it is not designed to do so). Then the question is what is the significance of the results in this paper? If an RL algorithm was not designed to maximize expected reward then what’s the use of proving it does not maximize expected reward?

---

> > > ### Author Response · Authors · 2022-08-07
> > > **Thanks for clarifying the concerns. We have revised the paper.**
> > >
> > > Dear reviewer,
> > >
> > > We greatly appreciate the response, and the clarifications of the concerns. These clarifications and actionable suggestions are important, as they allow us to better revise the paper to address these concerns. We have revised the paper to try to address all of the questions about the problem statement (changes highlighted in blue), and we provide more discussion below. **Can the reviewer confirm that the revisions address the concerns about defining the problem statement, or provide some feedback on further revising the paper to address the concerns?** We thank the reviewer for helping us improve the paper to better convey the main results.
> > >
> > > > So we have s state, a action but what are these variable part of (some state space, action space)? Are they integers (discrete)? Are they continuous on real-line?
> > >
> > > We have revised the Preliminaries (Section 2) to clarify these points. The states are elements of a state space $\mathcal{S}$ and actions are elements of an action space $\mathcal{A}$. Most of our analysis applies to both continuous and discrete state/action spaces, with the exception of Lemma 4.1; we have added a footnote to explain this.
> > >
> > > > What is $e$?
> > >
> > > We have also revised the Preliminaries to clarify this point. $e \in \mathcal{E}$ is a random variable indicating the task; $e$ can be either continuous (e.g., for an infinite number of tasks) or discrete. For example, if different tasks correspond to reaching different goal states, then the task $e$ is a goal state (i.e., $\mathcal{E} = \mathcal{S}$). This random variable $e$ is sampled $e \sim p_e(e)$ at the start of each episode, and the agent then receives the task-specific rewards $r_e(s_t, a_t)$ within this episode.
> > >
> > >
> > > > How does one define a task in MDP if not for the reward function? Why do I need anything other than a reward function to define a task?
> > >
> > > The reviewer is correct: only the reward function $r_e(s_t, a_t)$ is needed to define the task in the MDP.
> > >
> > > >  The description in paper 6 clears quite a few things. I would request the authors to please write the preliminary section so that the paper is self-contained.
> > >
> > > Double checking the definitions from [6], we were unable to find any missing definitions from our current paper, but would welcome suggestions in this regard! We have revised the description of OCBC to explain that this same method can be explained in different ways, and refer readers to [5] and [6] for alternative explanations, which provide some good intuition.
> > >
> > > > I do not see the difference between multi-objective RL and multi-task RL especially if the objective is to maximize expected reward.
> > >
> > > Thank you for clarifying this connection! We have revised the Preliminaries section to mention that multi-task RL and multi-objective RL are the same in this case, and to cite some prior work on multi-objective RL.
> > >
> > > > Then the question is what is the significance of the results in this paper? If an RL algorithm was not designed to maximize expected reward, then what’s the use of proving it does not maximize expected reward?
> > >
> > > We believe that most (if not all) OCBC methods _are_ designed to maximize expected reward. For example, [6] say that "[OCBC] methods … solve RL problems via conditional imitation learning." Similarly, [3] and [5] define their objective to be maximizing the expected sum of returns, and evaluate the proposed method using the expected sum of returns. _We believe that the results in the paper are significant because they show that prior methods designed for reward maximization can fail to maximize rewards._  We believe that this is useful not just because it highlights a failure case of prior methods, but also because it provides the mathematical machinery (Section 3) for relating OCBC methods to reward maximization, machinery that suggests that a small modification to these prior methods does provide a method that guarantees reward maximization.

---

> > > > ### Comment · Reviewer_MVgi · 2022-08-09
> > > > **response**
> > > >
> > > > Thanks for improving the preliminaries. I have improved my scores but I still feel uncertain regarding what is the relation and impact of these results to multi-objective RL (since now we agree it is the same here). And again in general difference between multi-task RL and multi-objective RL (and corresponding methods)

---

> > > > > ### Author Response · Authors · 2022-08-09
> > > > > **Thanks for the update!**
> > > > >
> > > > > Dear Reviewer,
> > > > >
> > > > > Thanks for the update!
> > > > >
> > > > > > relation and impact of these results to multi-objective RL
> > > > >
> > > > > We agree that fully understanding these connections is an interesting question. One potential impact of these results is that they might suggest how ideas from multi-objective RL might be used to build even better OCBC methods. For example, while OCBC methods typically sample the task from a fixed distribution at the start of each episode, more intelligent ways of sampling the tasks (e.g., related to CCS used in multi-objective optimization [1, 2]) might increase the sample efficiency.
> > > > >
> > > > > ----------------
> > > > >
> > > > > [1] Mossalam, Hossam, et al. "Multi-objective deep reinforcement learning." arXiv preprint arXiv:1610.02707 (2016).
> > > > >
> > > > > [2] Alegre, Lucas Nunes, Ana Bazzan, and Bruno C. Da Silva. "Optimistic linear support and successor features as a basis for optimal policy transfer." International Conference on Machine Learning. PMLR, 2022.

---

> > > ### Author Response · Authors · 2022-08-09
> > > **Do the revisions address the concerns?**
> > >
> > > Dear Reviewer,
> > >
> > > We wanted to reach out to see if our most recent reply and paper revisions have addressed the concerns. With only 1 day left in the rebuttal period, we were hoping that the reviewer could confirm that the revisions have addressed the concerns about the problem statement, further clarify the concerns, and/or provide actionable suggestions for further improving the paper. We would be happy to further revise the paper to clarify any concerns, or continue the discussion regarding any remaining concerns.
> > >
> > > We thank the reviewer for all their help so far in improving the paper!

---

> ### Author Response · Authors · 2022-08-05
> **Have the revisions addressed the reviewer's concerns?**
>
> Dear Reviewer,
>
> Thank you for raising a number of questions and concerns in the initial review. We understood the main concern of the review to be about the problem statement, a concern which we tried to address in our response below. **Has this response clarified the problem statement?** If not, we would be happy to provide alternative explanations of the problem statement, or to provide clarifications on any other part of the paper.

---

### Official Review · Reviewer_qZCX · 2022-07-11

**Rating:** 7
**Confidence:** 4
**Soundness:** 4 excellent
**Presentation:** 4 excellent
**Contribution:** 3 good

**Summary:**

The authors theoretically prove that the recent outcome-conditioned imitation learning methods do not always improve the policy. They analyze the OCBC methods from the reward maximizing view, which is the fundamental goal of imitation learning. And they find that the OCBC is a type of EM policy search and the average step makes it fail to find the optimal policy. Then they propose Normalized OCBC to "cancel out" the average step by importance weighting. The experimental results show that OCBC indeed fails to converge to the policy in practice and the proposed solution works well.

**Questions:**

Please see the Weaknesses part above.

**Limitations:**

The authors adequately addressed the limitations and potential negative societal impact of their work.

**Strengths And Weaknesses:**

Strengths:

The paper studies a fundamental problem of outcome-conditioned imitation learning. The theory is sound and the proposed method is simple and effective. The writing is good and I enjoy reading the paper.

Weaknesses:

1. Would you please introduce some failure cases in the real-world application of OCBC? in Line171, the failure condition "the tasks have different optimal actions but do visit similar states" is a little too demanding. I worry that in real-world settings, if you collect a diverse enough dataset, this phenomenon may never happen, which will limit the value of this paper.
2. For comparison results on benchmark tasks, there is only one baseline method. Although I don't think it is a big deal, it would be better to compare with more imitation learning methods.
3. The proposed Normalized OCBC is quite similar to some debiasing methods [1,2] in deep learning literature, i.e., using a biased model to cancel out the biasing effect. It is worthwhile to mention them in related work.

[1] Cadene, Remi, et al. "Rubi: Reducing unimodal biases for visual question answering." Advances in neural information processing systems 32 (2019).

[2] Bahng, Hyojin, et al. "Learning de-biased representations with biased representations." International Conference on Machine Learning. PMLR, 2020.

---

> ### Author Response · Authors · 2022-07-29
> **Response to qZCX**
>
> We thank the reviewer for their time reviewing our paper, and for the suggestions for improvement. We are glad to hear they enjoyed reading the paper.
>
> > Would you please introduce some failure cases in the real-world application of OCBC?
>
> Here's one hypothetical example where OCBC would fail in a real-world application:
> *Imagine you've collected hundreds of hours of data of a robot picking up a cup. Now, you want to teach the robot to slide the cup across a table. To learn this task, you use OCBC together with all the existing data. Using OCBC, you're most likely to find that the robot has learned to pick up the cup and place it at the other end of the table, rather than sliding the cup across the table.*
>
> >  in Line171, the failure condition "the tasks have different optimal actions but do visit similar states" is a little too demanding
>
> This sentence was meant as an example, and it is not the only condition under which OCBC will fail. In general, OCBC fails when the averaging step has a big effect. The averaging step will have a big effect if policy $\pi(a \mid s, e)$ visits a state where its policy is very different from the average policy, $\pi(a \mid s) = \int \pi(a \mid s, e') p(e' \mid s) de'$ (Footnote 1). Note that this average policy is a mixture of all the other policies, but weighted by how often they visit that state. If none of the other policies visit this state, then the average policy will just be the same as $\pi(a \mid s, e)$, and the averaging step won't have any effect. We will clarify this in the final paper.
>
>
> > I worry that in real-world settings, if you collect a diverse enough dataset, this phenomenon may never happen, which will limit the value of this paper.
>
> Simply collecting more diverse data will not necessarily fix OCBC. Similarly, the problems with OCBC occur even in the limit of infinite-capacity models with infinite data. For example, the bandit example (Fig 2) corresponds to collecting data uniformly from all tasks, and the failure to converge occurs even if an infinite amount of data is collected (i.e., it happens in the absence of sampling and approximation errors).
>
> >  there is only one baseline method. It would be better to compare with more imitation learning methods.
>
> As noted in L71, the baseline (OCBC) has been proposed in many prior works, so it refers to the method independently proposed by many prior papers. We will add citations to the other methods that propose this baseline in the figure captions (e.g., "GCSL," "RvS," "HID," "BC+HER").
>
> Imitation learning methods are not suitable for comparison because they make different assumptions than OCBC, which does not require expert demonstrations.
>
> > similar to some debiasing methods [1,2]
>
> We thank the reviewer for pointing us to these references, and will add them to the final version.
>
> **Have the responses above addressed the reviewer's concerns? Does the reviewer have any additional concerns?**

---

### Meta-Review · Area_Chair_12TY · 2022-08-26

**Recommendation:** Accept
**Confidence:** Certain

**Metareview:**

Paper shows that outcome-conditioned behavior cloning (OCBC) is not guaranteed to maximize long-term reward in multi-task RL. A simple, but effective variation of OCBC is proposed that does guarantee policy improvement.

While the scope of the paper is rather specific (a certain form of behavior cloning in multi-task RL), this family of methods has gained some momentum recently, while there are still many theoretical questions around it. This paper addresses some of these in a clear way and proposes a specific improvement. A clear accept in my view.

**Award:**

No

---

### Decision · Program_Chairs · 2022-09-14

Accept